# The impact of castration on physiological responses to exertional heat stroke in mice

**Christian K. Garcia, Gerard P. Robinson, Bryce J. Gambino, Michael T. Rua, Orlando Laitano, Thomas L. Clanton** [ID] *

Department of Applied Physiology and Kinesiology, University of Florida, Gainesville, FL, United States of America

* tclanton@hhp.ufl.edu

**Data Availability Statement:** All data for this manuscript is available in a data repository https://original-ufdc.uflib.ufl.edu//IR00011810/00001.

## Abstract

### Introduction

The capability of male mice to exercise in hot environments without succumbing to exertional heat stroke (EHS) is markedly blunted compared to females. Epidemiological evidence in humans and other mammals also suggests some degree of greater vulnerability to heat stroke in males compared to females. The origins of these differences are unknown, but testosterone has previously been shown to induce faster elevations in core temperature during acute, passive heat exposure. In this study, we tested the hypothesis that loss of testosterone and related sex hormones through castration would improve the performance and heat tolerance of male mice during EHS exposure.

### Methods

Twenty-four male mice were randomly divided into 3 groups, untreated EHS mice (SHAM-EHS), castrated EHS mice (CAS+EHS) and naïve exercise controls (NAIVE). Exercise performance and physiological responses in the heat were monitored during EHS and early recovery. Two weeks later, blood and tissues were collected and analyzed for biomarkers of cardiac damage and testosterone.

### Results

Core temperature in CAS+EHS rose faster to 39.5˚C in the early stages of the EHS trial (P<0.0001). However, both EHS groups ran similar distances, exhibited similar peak core temperatures and achieved similar exercise times in the heat, prior to symptom limitation (unconsciousness). CAS+EHS mice had ~10.5% lower body mass at the time of EHS, but this provided no apparent advantage in performance. There was no evidence of myocardial damage in any group, and testosterone levels were undetectable in CAS+EHS after gonadectomy.

### Conclusions

The results of these experiments exclude the hypothesis that reduced performance of male mice during EHS trials is due to the effects of male sex hormones or intact gonads.

**Funding:** U.S. Department of Defense, Medical Research and Materiel Command BA180078 (T.L. C.). The funders had no role in study design, data collection and analysis, decision to publish, or preparation of the manuscript.

**Competing interests:** The authors have declared that no competing interests exist.

However, the results are consistent with a role of male sex hormones or intact gonads in suppressing the early and rapid rise in core temperature during the early stages of exercise in the heat.

## Introduction

Exertional heat stroke (EHS) is characterized by profound central nervous system dysfunction and organ/tissue damage related to the combination of the high metabolic demands and large elevations in core temperature [1–3]. Long term consequences of EHS include higher incidence rates of cardiovascular events, chronic renal disease and other chronic disorders in survivors [4–6]. Incidences of multi-organ dysfunction and detrimental consequences of EHS can occur even with rapid cooling and appropriate post-heat stroke management [4, 6–8].

In the active members of the US Military service, heat illness incidence is similar among males and females [3]. However, when examining the most severe heat illness outcome, i.e. heat stroke, the majority of studies, both in and outside the military, are consistent with higher incidence rates in males [3, 9], though there is heterogeneity in results [10] and questions remain regarding whether this is behavioral or physiological in origin [11]. These differences in susceptibility to heat stroke are not limited to humans; e.g. in dogs, heatstroke diagnoses are much more frequent in males than females [12].

Our laboratory has developed a preclinical model of EHS in mice that has been useful in studying the sex differences in response to exercise in the heat [10]. Throughout this report, we use the abbreviation, EHS, for both the clinical condition in humans and the response of mice to our experimental EHS model. Using this preclinical model, we previously demonstrated that female mice have a greater capacity for exercising in the heat, reaching greater power output, longer times of exposure, and attaining higher maximum running speeds. However, both sexes succumb to similar neurologic dysfunction at nearly identical peak core temperatures ($T_{c,max}$) [13, 14]. The underlying cause of these differences in response is not well understood [14, 15]. Another sex difference that we have observed is that females develop a metabolic disorder in the heart following EHS that emerges only after 9–14 days of recovery that is not seen in males [16]. These differences in both performance and later outcomes likely reflect, in part, the impact of sex hormones originating from male and/or female gonads, though they can also arise from sex chromosome differences that can impact gene expression [17].

Testosterone is a potent androgenic hormone with a wide range of biological effects, including stimulation of protein synthesis and inhibition of protein degradation [18]. This increase in anabolic activity contributes to larger total body mass and lean mass seen in adult male mice (for C57BL/6, ~29% greater total mass in males:~25% greater lean mass [19]). In addition, with increases in body mass, there is a reduction in body mass/ surface area used for heat dissipation [20]. In a previous study, body mass, sex, and exercise power output were strong predictors of performance during the EHS protocol in male and female mice when predicting the onset of the symptom-limited endpoint (i.e., neurological dysfunction) [14].

In animal models, testosterone has been shown to also seasonally affect thermoregulation, within interspecies competition, under stress related conditions, and in terms of overall activity [21, 22]. In the Afrotropical pouched mouse (*Saccostomus campestris; rodentia*), testosterone inhibited torpor in males, increasing their overall basal temperature 4°C higher than females [23]. In rats, the loss of testosterone also results in a modification in the responses to

chronic heat exposure. For example, castrated rats exposed to eight weeks of heat exposure had slower growth rates, decreased resting core body temperature, and increased plasma corticosterone. Administration of testosterone reverted these parameters, suggesting a strong role of testosterone on heat balance regulation [24]. In mice, castration dampens the thermoregulatory response to acute heat exposure, an outcome reversed by testosterone supplementation [25]. Because of the extensive effects of testosterone on temperature regulation and metabolic activity, we hypothesized that in a preclinical model of EHS, testosterone loss by castration would lead to a suppressed or delayed rise in core body temperature during exercise, providing "protection" from the symptom-limited endpoint and allowing for better overall performance in the heat. This could explain our previous observations regarding the differences in EHS responses between males and females [14].

## Methods

### Animal subjects

This study was approved by the University of Florida's Institutional Animal Care and Use Committee and met ARRIVE 2.0 Guidelines. All male mice (n = 24) were C57BL/6J (Jackson Laboratories, Bar Harbor, ME) and upon arrival, housed on a 12:12-h light/dark cycle at 20°C to 22°C and 30% to 60% relative humidity (RH). Standard chow (2918 Envigo; Teklad, Madison, WI) and water were provided ad libitum. All mice were approximately 10 weeks old upon arrival and were approximately 18 weeks old at the end of the study. Mice were the experimental unit and were randomly allocated by the authors into either a castration + exertional heat stroke (CAS+EHS) group or an exertional heat stroke only group (SHAM-EHS). The SHAM-EHS group received all treatments, including abdominal surgery, but castration. Another control group (NAIVE) was studied precursory to the previous groups but the mice were of the same age, underwent the same exercise training protocol, were not castrated, and did not undergo the EHS protocol. Their inclusion was to differentiate the effect of heat on the biochemical analyses. Fourteen days after the EHS protocol, tissue collection and plasma collection were performed on all mice for biochemical analyses and the mice were euthanized.

### Castration, emitter implantation, and exercise training

Under 2.5% isoflurane anesthesia at a rate of 0.6 L/min, the abdomen was prepared for surgery using aseptic technique. All mice were implanted with a temperature emitter (G2 E-Mitter; Starr Life Sciences, Oakmont, PA) as previously described [26]. The same surgical opening was used to perform castration (CAS). For CAS, a sterilized forceps was used to reach into the peritoneal cavity and retrieve the testicular fat pads. Using a cautery pen, the testicular artery was cauterized along with the fat pad. Once severed by cauterization, the entire testis and associated fat pad were removed. The process was repeated for the other testis. The abdominal cavity was closed, and mice were singly housed during recovery and for the rest of the experiment. They were monitored for pain and behavior for 48 h after surgery, and subcutaneous injections of buprenorphine were given every 12 h for analgesia during this period. Mice recovered from surgery for 2 weeks and then were given modified in-cage running wheels (model 0297–0521; Columbus Instruments, Columbus, OH) to allow for voluntary exercise training for 3 weeks. During the third week of voluntary running, animals were exercised on a forced running wheel on four occasions over four consecutive days (Lafayette, model 80840, Lafayette, IN; powered by a programmable power supply). Once training was completed, mice were given 2–3 full days of rest before the EHS trial. In-cage running wheels were made available during that time. The total time for recovery from castration exceeded 5 weeks.

## EHS protocol

The EHS protocol was carried out as previously reported [13, 26]. Mice were brought to the laboratory the afternoon before EHS. Core temperature (Tc) was recorded in 30-s intervals throughout the night to ensure normal temperature profiles before EHS. The 12:12-h light/dark cycle was maintained. The EHS procedure was run in the early morning beginning at approximately 7:30 AM. Mice remained in their cages with Tc being monitored while the environmental chamber (Thermo Forma, 3940; Thermo-Fisher, Waltham, MA) heated to a set point of 37.5°C, and 30–40% relative humidity. Environmental chamber temperature and humidity were measured and recorded at the location of the running wheel. Once the temperature equilibrated (30–45 min), the mice were placed in the enclosed forced running wheel within the chamber. Mice were given >5 min to recover from the stress of being handled and then, once environmental temperature stabilized again to < 37.5°C, the running wheel was started on a preprogrammed and standardized incremental protocol (Fig 1). Speed began at 3.1 m•min$^{-1}$ and increased 0.3 m•min$^{-1}$ every 10 min until the mouse reached a Tc of 41°C. Once 41°C Tc was reached, a "steady-state" exercise phase began with speed maintained until the symptom-limited end point. The EHS end point for this model was previously defined by loss of consciousness, specifically, three consecutive revolutions of the wheel with no physical responses by the mouse. Upon symptom limitation, mice were checked for responsiveness to tactile touch and then quickly removed from the exercise wheel and placed in their cage with ad libitum access to food and water. The environmental chamber door was opened to room air during the recovery period and the temperature in the incubator rapidly returned to room temperature. Tc was recorded continuously until 3 h post-EHS. The animals were returned to the vivarium for two weeks of recovery.

## Sample collection and biochemical analyses

At 14 days post-EHS mice were placed under isoflurane anesthesia, whole blood samples were drawn via transthoracic cardiac stick, using a 27-gauge needle, preloaded with ethylenediaminetetraacetic acid to prevent clotting. Animals were then euthanized under deep isoflurane anesthesia by excision of the heart. Whole blood was then centrifuged at 4°C for 10 min at 2000g, the plasma transferred to microtubes and immediately snap frozen in liquid nitrogen. Biochemical analyses were done for cardiac troponin T, bile acids and for testosterone.

## Quantification of total bile acids

Because we have previously identified a metabolic disorder in the heart in female mice, 2 weeks after this model of EHS, at the end of the study, we evaluated several biomarkers for heart disease in these male mice, including circulating bile acids, which is a link between gut injury, typical response to EHS and heart effects [27] and Troponin T [28], an indicator of heart injury. Bile acids were measured from plasma using the Mouse Total Bile Acids Assay, (Crystal Chem; Illinois,USA Catalog #80471). Triplicate plasma samples were analyzed in a 96 well microplate. In the presence of NAD, bile acids converted into 4-keto steroids and NADH. The NADH reacted with nitrotetrazolium blue to form blue color. Dye formation was monitored via absorbance at 540 nm, using a spectrophotometer.

## Quantification of Cardiac Troponin T

Cardiac Troponin T was determined using an ELISA kit (Biomatik; Delaware, USA Catalog #EKU07910). The double antibody sandwich microtiter plate provided in this kit was pre-coated with an antibody specific to Cardiac Troponin T. Standards or samples were added to

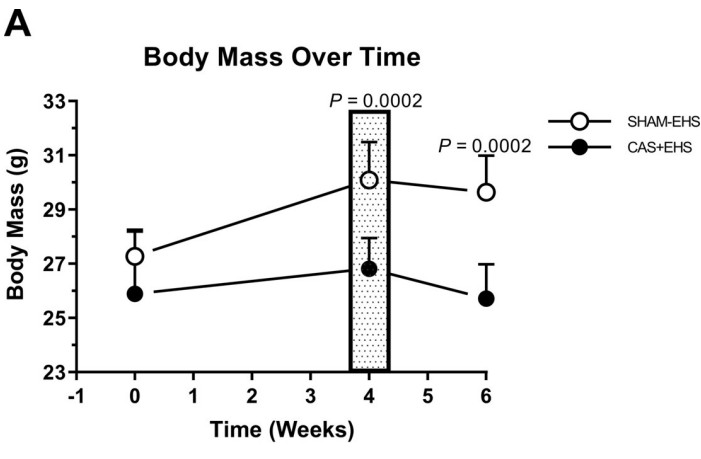

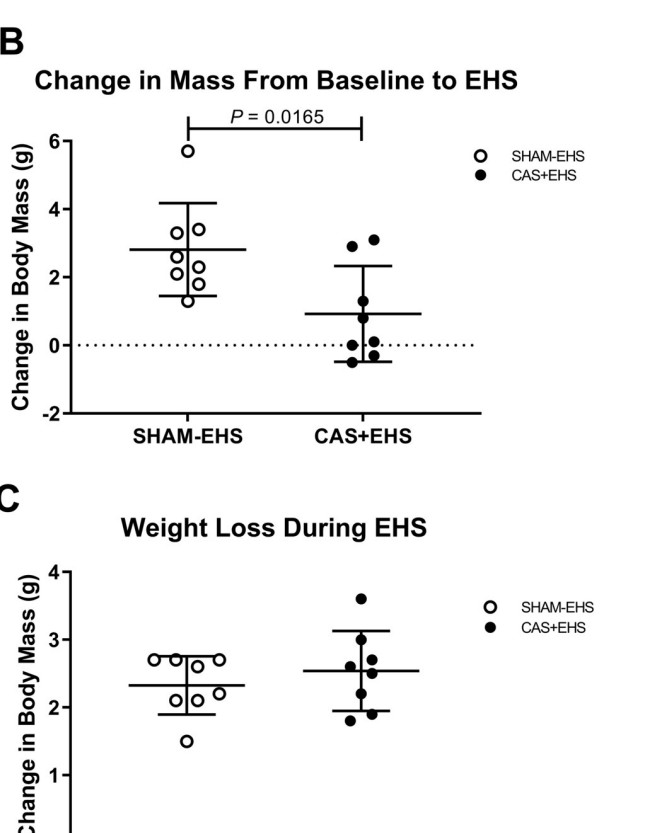

**Fig 1. A)** Timeline of changes in body mass throughout experiment. CAS+EHS exhibited lower weights after baseline compared to SHAM-EHS mice. Patterned box at week 4 represents the time within the protocol the mice were exposed to EHS. *P* values are Bonferroni corrected; Effect sizes: 0.83, 2.74, 3.19, respectively across time. ANOVA followed by orthogonal t-test comparisons between groups at each time point. **B)** Loss of body mass due to castration from baseline to pre EHS protocol. T- test, Effect size: 1.35. **C).** Loss of mass during the EHS protocol, not significant, t-test. Effect size: 1.1. All data, all data are means ± SD; n = 8 per group.

the appropriate microtiter plate wells with a biotin-conjugated antibody specific to Cardiac Troponin T. Next, Avidin-conjugated to Horseradish Peroxidase was added to each microplate well and incubated. After TMB (3,3'5,5'-tetramethylbenzidine) substrate solution was added,

only those wells that contain Cardiac Troponin T, biotin-conjugated antibody and enzyme-conjugated avidin exhibited a change in color. The enzyme-substrate reaction was then terminated by the addition of sulfuric acid solution and the color change measured spectrophotometrically at 450 nm.

## Quantification of testosterone levels

A Multi-Species Hormone Magnetic Bead Panel (MSHMAG-21K, Millipore,Burlington, MA) was used to determine testosterone levels in CAS+EHS and EHS mice using multiplex Luminex technology. Triplicate plasma samples were prepared by adding acetonitrile to the samples for 10 min. Samples were then centrifuged at 17,000 x g for 5 minutes. The supernatant was collected and placed into microcentrifuge tubes. Supernatant samples were desiccated in a vacufuge for 1.5 h at the highest vacuum setting. Afterwards samples were reconstituted in the kit's assay buffer and immediately assayed.

## Thermal area calculation

Ascending thermal area was calculated from the serial, real time measurements of Tc using the following formula:

$$Thermal\ area = \int_{t_{start}}^{t_{end}} (Tc(t) - 39.5)dt$$

Where:

Tc (t) = core temperature at time (t), above 39.5°C

$t_{start}$ = time at beginning of forced running

$t_{end}$ = time at symptom limitation, $T_{c,max}$

This calculation was used to determine the thermal load above 39.5°C. This threshold temperature was used since EHS has been shown to induce damage at lower thermal loads than passive heat stroke [29], and it has originally developed for a passive heat stroke model [30] but has since been used extensively for EHS models in the mouse [29].

## Statistical and analytical approach

Statistical testing and graphics were performed using SAS-JMP Pro 15 and/or GraphPad Prism 9.4. G*Power 3.1 was used to calculate effect size and the minimum sample size needed for the experiment's primary outcome variables of 'exercise time' in the heat and 'ascending thermal area' obtained from previous research [14]. Using effect size of d = 1.3 for both exercise time and ascending thermal area and β-1 = 0.8, α = 0.05, for two tailed tests, a minimum sample size of 8 in each group was determined to be necessary. For all analyses, normality of groups were determined using the Anderson-Darling goodness of fit test. All comparisons of parametric data with similar variance were performed using the two-sample t-test. Non-parametric data were compared with the Kruskal-Wallis test for three groups; effect size was tested based on ref. [31]. For two group comparisons of non-parametric data, the Mann-Whitney U test was used. For both student t-tests and Mann-Whitney U test. For body weight changes over time between groups, ANOVA was utilized followed by limited orthogonal t-test comparisons, restricted to differences in body weight between groups at three specific time points, with corrections for false discovery rate using Bonferroni's test.

## Results

### Differences in physical measurements and performance

Before castration and emitter implantation, CAS+EHS mice had similar body mass compared to SHAM-EHS mice (Fig 1A). At 4 weeks, post-surgery, CAS+EHS mice were lighter than SHAM-EHS mice with an average weight gain of only +0.92 ± 0.50 g for CAS+EHS mice compared to +2.80 ± 0.48 g for the SHAM-EHS mice (Fig 1B). This difference in weight between groups persisted through the entire 14 days of recovery from EHS (Fig 1A).

In terms of performance, CAS+EHS and SHAM-EHS mice ran approximately the same absolute distances and for similar total time periods (Fig 2A). Interestingly, CAS+EHS mice exhibited faster elevations in core temperature in the early stages of the EHS trial, requiring less time to reach 39.5˚C ($P<0.0046$) (Fig 2A and 2B). However, the CAS+EHS mice did not differ from SHAM-EHS mice in terms of the absolute time required for core temperature to become elevated from 39.5˚C to 41˚C or from 41˚C to $T_{c,max}$.

Absolute ascending thermal area, the distance run and the total time to reach $T_{c,max}$ were not different between CAS+EHS and SHAM-EHS mice (Fig 3A–3C).

Body surface area (BSA; calculated from body mass using the Meeh's equation [20]) was smaller in CAS+EHS mice compared to SHAM-EHS mice ($P<0.0002$) (Fig 4A) and thus BSA/m (body surface area to mass ratio) was larger ($P<0.0001$) (Fig 4B). This means that even though the mice were a smaller body mass and greater body surface/mass ratio, they received no advantage from this in terms of performance in the heat or resistance to heat stroke.

### Biochemical analyses

Plasma cardiac troponin T did not differ in NAIVE, CAS+EHS or SHAM-EHS mice at the end of 14 day of recovery (Fig 5A). Total bile acids also did not differ between NAIVE, CAS+EHS or SHAM-EHS mice (Fig 5B). Plasma testosterone levels were undetectable in the CAS+EHS mice when compared to their SHAM-EHS counterparts ($P<0.002$) (Fig 5C).

## Discussion

Castration did not have a significant impact on the absolute levels of exercise performance in the heat or resistance to acquiring EHS in this preclinical model of EHS. Although castrated and intact mice ran to roughly equivalent distances and times in the heat, castrated mice were carrying ~7% less mass at the time of the EHS trial. Based on previously described linear relationships between body mass and distance run in this model [14], the castrated male mice were projected to run an average of ~125 additional meters in the heat, which would have extended their EHS trial by ~25 minutes. This is roughly how much the mean value changed for these variables in the castrated group, though it did not reach statistical significance. Therefore, the performance of the intact mice in the heat was not noticeably or significantly affected by available testosterone or other factors related intact gonads and was independent of changes in mass.

These results are in contrast to previous reports in mouse models of passive heat stroke that report that castration results in slower and less severe elevations in core temperature during exposure to passive hyperthermia [25] and marked improvements in thermal tolerance and resistance to passive heat stroke [32]. However, the physiology of passive and exertional heat stroke can be quite different, with the effects of EHS having more predominant influences on rhabdomyolysis, kidney dysfunction, hypokalemia, and hypoglycemia [1]. Furthermore, in EHS, heat tolerance during exercise in the heat is likely to be more dependent on underlying

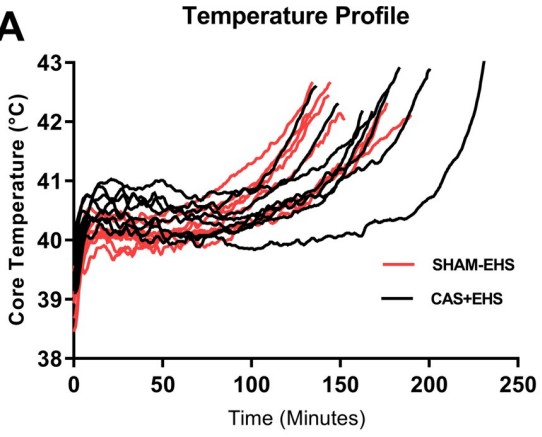

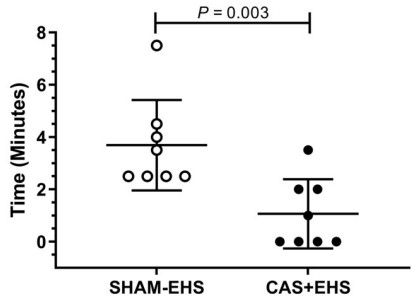

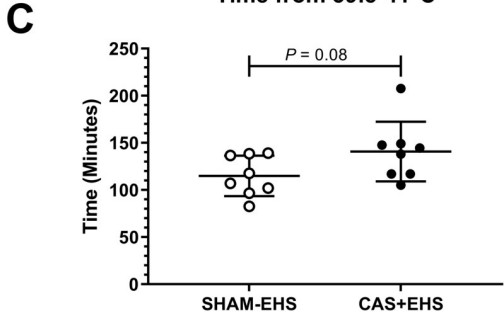

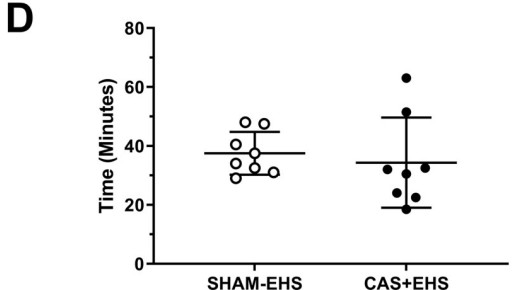

**Fig 2. A)** Core temperature-time profile during the EHS runs, comparing all CAS+EHS (black) and SHAM-EHS mice (red). **B)** Comparison of the time from the start of EHS protocol to Tc = 39.5°C, (Mann Whitney U test), Effect size: 1.71. C) Comparison of exercise time from 39.5°C to 41°C, between CAS+EHS or SHAM-EHS mice (t-test). Effect size: 0.96 **D)** Comparison of time from 41°C to Tc,max. Effect size: 0.34 Means ± SD; n = 8 per group. All data are means ± SD.

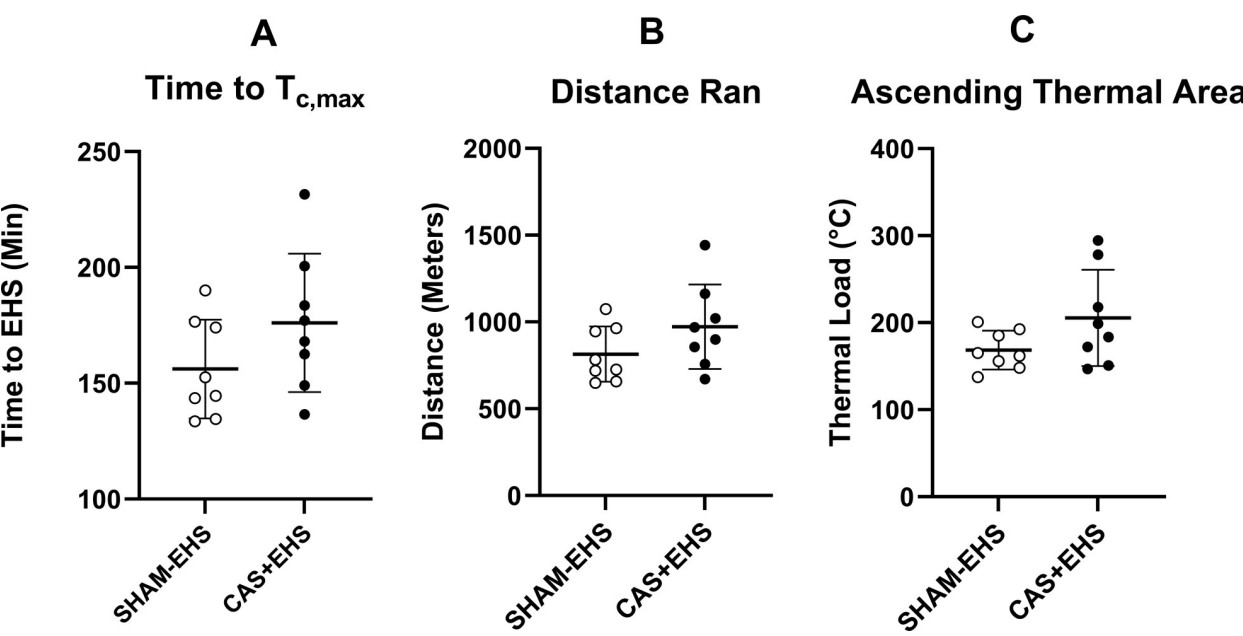

**Fig 3. A)** The total time taken for mice to reach the symptom-limited endpoint, $T_{c,max}$, of the EHS trial. Effect size: 0.77. **B)** The total distance run by the end of the EHS trial for CAS+EHS and SHAM-EHS mice. Effect size: 0.76. **C)** The cumulative ascending thermal area of EHS during the EHS trial (defined by the integration of Tc >39.5˚C and time. Effect size: 0.89. All Data: means ± SD; (t-test comparisons), n = 8 per group.

cardiovascular and exercise conditioning. So direct comparisons between outcomes of passive and exertional heat stroke need to be viewed with caution.

Another outcome of this study was the lack of evidence for emerging heart abnormalities in castrated mice, at least based on the absence of changes in bile acids or troponin T in the circulation, two weeks after EHS. We previously observed severe metabolic and inflammatory disturbances in female ventricles that emerged only after 9–14 days of recovery that was not seen in male mice [16]. Though the type of data collected here is not a direct comparison with the

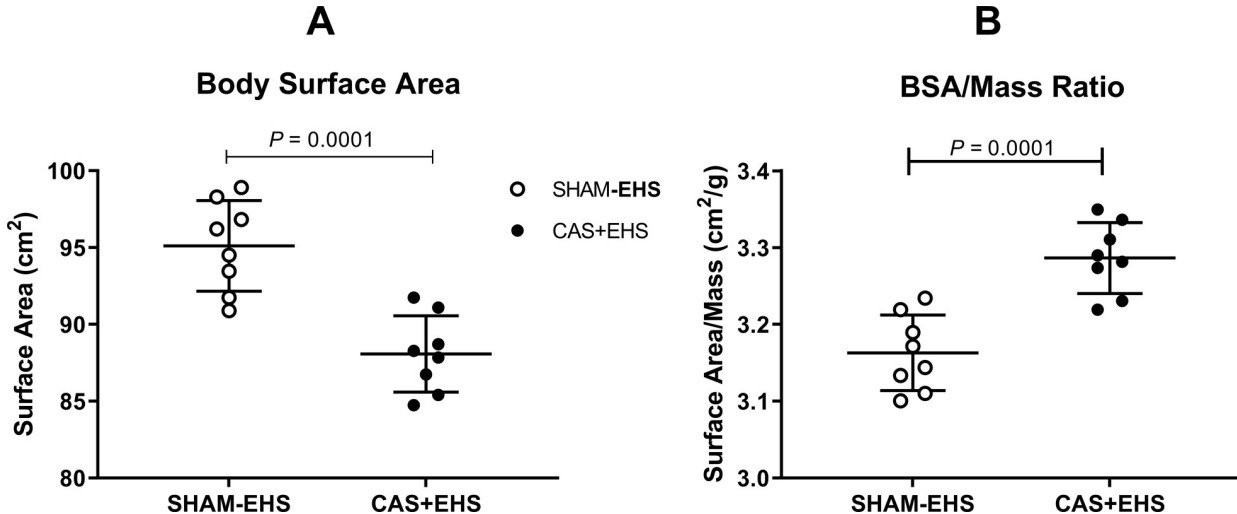

**Fig 4. A)** Body surface area was calculated by using Meeh's equation [17]. Effect size: 2.54. **B).** Body surface area to mass ratio. Effect size: 2.1 Means ± SD, n = 8, (t-test comparisons).

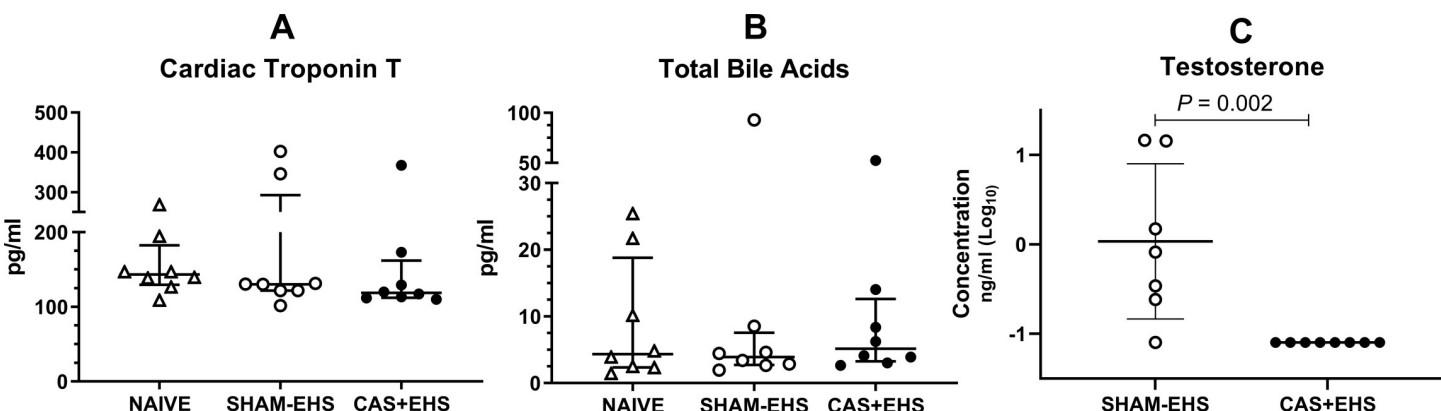

**Fig 5.** A) Comparison of Cardiac Troponin T. No differences were detected in plasma samples between NAIVE (wild type-no EHS), SHAM-EHS, and CAS+EHS mice; effect size: 0.007 (Kruskal-Wallis test). B) Comparison of Total Bile Acids, (Kruskal Wallis); effect size: -0.07. C) Comparison of testosterone levels (Mann Whitney U test); effect size:1.84. Samples were measured in plasma using the MAGPIX Luminex platform. Means±SD, n = 8 for Sham and CAS+EHS; n = 7 for SHAM-EHS.

previous report, the continued absence of evidence of a pathological response in both control and castrated males is consistent with the idea that lack of testosterone does not seem to be primarily responsible for myocardial vulnerability in females seen in the two weeks following EHS that were previously reported [16].

One interesting and unique finding was that castrated mice exhibited a more rapid rise to 39.5˚C in the early stages of the EHS protocol, compared to their naïve counterparts. This outcome is in direct contrast to previous reports that castration in male C57BL/6 mice can suppress the early hyperthermic response when exposed to passive heat [25], an outcome reversed by testosterone supplementation. As the EHS protocol progressed, other factors of thermoregulation must have predominated as both groups in this study tended to have relatively similar core temperature profiles over the remainder of the heat exposure. We do not have an explanation for these differences in findings between studies, but it must relate to the fact that our mice were exercising in hyperthermia, bringing out other integrative physiological responses that were not present in passive exposure to hot environments.

By what mechanisms could sex hormones affect thermoregulation or exercise tolerance in the heat? Both estrogen and testosterone are vasodilators, stimulating the release of nitric oxide (NO) and upregulating endothelial nitric oxide synthase (eNOS) in the vasculature endothelium [33, 34]. Neuronal nitric oxide synthase (nNOS) is also upregulated by estrogen exposure [35] but testosterone or dihydrotestosterone have no apparent effects on its expression or activity, at least in the endothelium [34]. The bioavailability of NO in the vasculature is a major factor in the normal control of endothelial-dependent vascular regulation [36], and NO is a necessary component of active vasodilation in heat stress in humans [37]. Women are shown to have a greater overall vasodilatory response than males, a response that can be partially inhibited by blocking NOS activity [38]. Though some have argued for a role of NO in the rapidly developing hypotension and cardiovascular collapse near the end of heat stroke [39], other investigators have demonstrated that NO plays essentially no substantial role in the hypotension associated with the late stages of heat stroke progression [40].

Sex hormones also play important roles in immune cell regulation. Estrogen promotes immune defense, whereas testosterone and progesterone are known to suppress certain elements of the inflammatory response [14, 41, 42]. Immune regulation is an important component of the responses to hyperthermia, particularly for an organism to recover from the damaging effects of hyperthermia [43]. However, any role of the immune system during the

acute exposure to exercise in the heat seems doubtful due to the actions and timing of the immune cytokines released [14, 43].

Sex hormones also affect exercise performance and muscle function, which could directly impact performance in this model. When used as a supplement during exercise in male mice, estrogen increases exercise performance on the treadmill [44], and the greater exercise capacity in female mice is abolished following ovariectomy [44]. Besides its well-known androgenic effects on protein synthesis, acute administration of testosterone also appears to improve muscle performance, increasing strength and endurance, particularly in fast-twitch muscles [45, 46]. Another potential factor is that female mice are endowed with a greater proportion of slow oxidative fibers in limb muscles [47], which are well-suited for endurance exercise, lowering energy costs and presumably allowing for greater cardiopulmonary reserve during the EHS protocol.

In summary, our results appear to exclude one important factor, namely, male sex hormones, that might have explained the large sex differences in performance during exertional heat stroke in mice we previously reported [14]. We conclude that testosterone, dihydrotestosterone, or other factors related to intact gonads, do not increase risk in dealing with exercise in exertional heat stroke. In fact, it is likely that testosterone plays a protective role. Intact gonads in male rats have been shown to be necessary for a normal Hsp70 expression in the male heart in response to intense exercise [48]. Therefore, testosterone or other sex hormones likely support the response to thermal stress at a variety of physiological levels. Our results hold clinical relevance, not only for understanding differences between sexes, but also in addressing the potential physiological effects of-, and increasing use of androgen replacement therapy and/or supplementation for performance, a topic of health concern to athletes [49] and to the military [50].

## Acknowledgments

The authors have no competing or conflicting interests related to this research. The opinions or assertions herein are the private views of the authors and are not construed as official or as reflecting the views of any other organization

## Author Contributions

**Conceptualization:** Christian K. Garcia, Thomas L. Clanton.

**Data curation:** Christian K. Garcia, Bryce J. Gambino.

**Formal analysis:** Christian K. Garcia, Orlando Laitano, Thomas L. Clanton.

**Funding acquisition:** Thomas L. Clanton.

**Investigation:** Christian K. Garcia, Gerard P. Robinson, Bryce J. Gambino, Michael T. Rua.

**Methodology:** Christian K. Garcia, Gerard P. Robinson, Bryce J. Gambino, Michael T. Rua.

**Project administration:** Thomas L. Clanton.

**Writing – original draft:** Christian K. Garcia.

**Writing – review & editing:** Christian K. Garcia, Gerard P. Robinson, Bryce J. Gambino, Michael T. Rua, Orlando Laitano, Thomas L. Clanton.

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
