## [Decision Letter · Decision Letter 0]

16 Jun 2022

PONE-D-22-14934The impact of castration on physiological responses to exertional heat stroke in mice PLOS ONE

Dear Dr. Clanton,

Thank you for submitting your manuscript to PLOS ONE. After careful consideration, we feel that it has merit but does not fully meet PLOS ONE’s publication criteria as it currently stands. Therefore, we invite you to submit a revised version of the manuscript that addresses the points raised during the review process.

We look forward to receiving your revised manuscript.

Kind regards,

William M. Adams

Academic Editor

PLOS ONE

Journal Requirements:

2. As part of your revision, please complete and submit a copy of the Full ARRIVE 2.0 Guidelines checklist, a document that aims to improve experimental reporting and reproducibility of animal studies for purposes of post-publication data analysis and reproducibility: https://arriveguidelines.org/sites/arrive/files/Author%20Checklist%20-%20Full.pdf (PDF). Please include your completed checklist as a Supporting Information file. Note that if your paper is accepted for publication, this checklist will be published as part of your article.

"The authors have no competing or conflicting interests related to this research. The research was supported by contracts from the U.S. Department of Defense, BA180078 (T.L. Clanton), with supplemental support from the B.K. and Betty Stevens Endowment at the University of Florida (T.L. Clanton). The opinions or assertions herein are the private views of the authors and are not construed as official or as reflecting the views of the Army or the Department of Defense. Citations of commercial organizations and trade names in this report do not constitute an official Department of the Army endorsement or approval of the products or services of these organizations"

 "U.S. Department of Defense, BA180078 (T.L.C.), U.S. Department of Defense, BA180078 (T.L.C)"

 "U.S. Department of Defense, BA180078 (T.L.C.), U.S. Department of Defense, BA180078 (T.L.C)"

Reviewers' comments:

Reviewer's Responses to Questions

**Comments to the Author**

1. Is the manuscript technically sound, and do the data support the conclusions?

Reviewer #1: Yes

Reviewer #2: Yes

2. Has the statistical analysis been performed appropriately and rigorously? 

Reviewer #1: Yes

Reviewer #2: Yes

3. Have the authors made all data underlying the findings in their manuscript fully available?

Reviewer #1: Yes

Reviewer #2: Yes

4. Is the manuscript presented in an intelligible fashion and written in standard English?

Reviewer #1: Yes

Reviewer #2: Yes

5. Review Comments to the Author

Reviewer #1: The purpose of this manuscript was to investigate the effects of castration, as a means of reducing endogenous testosterone, on physiological responses to exertional heat stress. The manuscript is well written and easy to follow. This reviewer has a few minor comments that may help with the clarity and reception:

Throughout the manuscript EHS is used to refer to both the protocol and the medical condition. It would be beneficial to clarify this acronym confusion.

It would be interesting to replicate these study results with an additional arm that examines both castration and exogenous estrogen/progesterone administration.

Abstract:

Line 1: The combination of “resilience of male mice to sustain” and “blunted” in the first sentence overly complicates this concept. Recommend revision.

Line 22: This conclusion statement is contradictory to your hypotheses and thus “consistent” may not be the best word for the latter part of this statement.

Introduction:

Line 37-39: It is not clear to this reviewer immediately how these statements differ, recommend clarifying.

Line 39: At best the literature on EHS incidence and sex differences is mixed. Please see: 10.1016/j.envres.2018.10.020, 10.4085/1062-6050-539-19. In this reviewer’s opinion it is better to reflect this variability, even if it contradicts the hypothesis.

Line 74: The concept of a decreased temperature during EHS is confusing to this reviewer. I think the authors may be referring to the exercise heat protocol rather than the condition itself. Suggest clarifying throughout the manuscript. It may be helpful to designate an acronym other than EHS for the protocol, as this is widely understood to represent exertional heat stroke as the medical condition.

Methods:

Line 88: Suggest clarifying whether the SHAM group was castrated.

Line 193: Please clarify the input metrics for this sample size determination.

Results:

Line 221: Specify the use of Meeh’s equation within this statement.

Figure 3: CAS mice appeared to have greater variability in responses than Naïve, could the authors comment on this observation within the discussion?

Line 252: Greater emphasis on the difference between the present data and the literature is needed. I.e., passive vs exertional models resulting in different responses.

Reviewer #2: In this manuscript, the authors detail an experiment intended to test whether castration, and therefore the removal of testosterone, from male mice affects the physiological response to EHS. Contrary to previous findings, no substantial differences were observed in the castrated mice compared to the control (EHS) mice. While the sample size is small, I believe the experimental design is adequate, and the conclusions/discussion are in concordance with the reported results. However, I have detailed a few comments that need to be addressed in the attached document.

6. PLOS authors have the option to publish the peer review history of their article (what does this mean?). If published, this will include your full peer review and any attached files.

Reviewer #1: **Yes: **Luke N. Belval, PhD

Reviewer #2: **Yes: **Aaron R. Caldwell

---

## [Author Response · Author response to Decision Letter 0]

8 Sep 2022

Response to Reviewers: Manu. D-22-14934. Garcia et al. “The impact of castration on…..”

We wish to thank the reviewers and editor for their careful reviews and their considerate and constructive comments. The manuscript has been improved by your input. 

Reviewer #1: The purpose of this manuscript was to investigate the effects of castration, as a means of reducing endogenous testosterone, on physiological responses to exertional heat stress. The manuscript is well written and easy to follow. This reviewer has a few minor comments that may help with the clarity and reception:

Thank you for your positive comments on the overall manuscript. 

Throughout the manuscript EHS is used to refer to both the protocol and the medical condition. It would be beneficial to clarify this acronym confusion.

With regard to the acronym, EHS, we are using the same acronym for both the clinical condition in humans and the preclinical model. To make this clear, we have included new text in lines 45-46. “Throughout this report, we use the abbreviation, EHS, for both the clinical condition in humans and the response of mice to our experimental EHS model.”

It would be interesting to replicate these study results with an additional arm that examines both castration and exogenous estrogen/progesterone administration.

We agree that the additional arms using exogenous estrogen and progesterone administration would be valuable. We are hoping to be able to design and complete such work in future experiments. 

Abstract:

Line 1: The combination of “resilience of male mice to sustain” and “blunted” in the first sentence overly complicates this concept. Recommend revision.

This has been changed to: Lines 1-2: “The capability of male mice to exercise in hot environments without succumbing to exertional heat stroke (EHS) is markedly blunted compared to females.”

Line 22: This conclusion statement is contradictory to your hypotheses and thus “consistent” may not be the best word for the latter part of this statement.

Introduction:

This has been changed to:

Lines 20-24: “The results of these experiments exclude the hypothesis that reduced performance of male mice during EHS trials is due to the effects of male sex hormones or intact gonads. However, the results are consistent with a role of male sex hormones or intact gonads in suppressing the early and rapid rise in core temperature during the early stages of exercise in the heat.”

Line 37-39: It is not clear to this reviewer immediately how these statements differ, recommend clarifying.

This has been changed to:

Lines 35-40. “In the active members of the US Military service, heat illness incidence is similar among males and females [3]. However, when examining the most severe heat illness outcome, i.e. heat stroke, the majority of studies, both in and outside the military, are consistent with higher incidence rates in males [3,9], though there is heterogeneity in results [10] and questions remain regarding whether this is behavioral or physiological in origin [11].

Line 39: At best the literature on EHS incidence and sex differences is mixed. Please see: 10.1016/j.envres.2018.10.020, 10.4085/1062-6050-539-19. In this reviewer’s opinion it is better to reflect this variability, even if it contradicts the hypothesis.

Thank you, as reported in the last comment, we have included these references and though they are largely in support of the original statement, we nevertheless agree that it is wise to soften this conclusion to reflect the current consensus. 

Line 74: The concept of a decreased temperature during EHS is confusing to this reviewer. I think the authors may be referring to the exercise heat protocol rather than the condition itself. Suggest clarifying throughout the manuscript. It may be helpful to designate an acronym other than EHS for the protocol, as this is widely understood to represent exertional heat stroke as the medical condition.

Thank you. To avoid confusion, we have added the following definition to the intro: “Throughout this report, we use the abbreviation, EHS, for both the clinical condition in humans and the response of mice to our experimental EHS model.”

Methods:

Line 88: Suggest clarifying whether the SHAM group was castrated.

We have made an adjustment to the terms used to avoid confusion, the EHS-sham group was not castrated. Lines 96-99: “Mice were randomly allocated into either a castration + exertional heat stroke (CAS+EHS) group or an exertional heat stroke only group (SHAM-EHS). The SHAM-EHS group received all treatments, including abdominal surgery, but without castration. Another control group (NAIVE) was studied precursory to the previous groups but the mice were of the same age, underwent the same exercise training protocol, were not castrated, and did not undergo the EHS protocol.” 

Line 193: Please clarify the input metrics for this sample size determination.

The following has been added:

Lines: 203-208: “G*Power 3.1 was used to calculate effect size and the minimum sample size needed for the experiment’s primary outcome variables of ‘exercise time’ in the heat and ‘ascending thermal area’ obtained from previous research [14]. Using effect size of d = 1.3 for both exercise time and ascending thermal area and �-1 = 0.8, �= 0.05, for two tailed tests, a minimum sample size of 8 in each group was determined to be necessary..”

Results:

Line 221: Specify the use of Meeh’s equation within this statement.

This has been changed to:

Line 236 and in figure legend 4: “Body surface area was calculated by using Meeh’s equation [20].”

Figure 3: CAS mice appeared to have greater variability in responses than Naïve, could the authors comment on this observation within the discussion?

When testing for variance, the O’Brien test for homogeneity of variance did not detect any significant differences. We do not feel that there was sufficient evidence to report greater variability in one group over the other for this variable, so we prefer not to comment. 

Line 252: Greater emphasis on the difference between the present data and the literature is needed. I.e., passive vs exertional models resulting in different responses.

The following has been reworded and added: 

Line 259-268: “These results are in contrast to previous reports in mouse models of passive heat stroke that report that castration results in slower and less severe elevations in core temperature during exposure to passive hyperthermia [25] and marked improvements in thermal tolerance and resistance to passive heat stroke [32]. However, the physiology of passive and exertional heat stroke can be quite different, with the effects of EHS having more predominant influences on rhabdomyolysis, kidney dysfunction, hypokalemia, and hypoglycemia [1]. Furthermore, in EHS, heat tolerance during exercise in the heat is likely to be more dependent on underlying cardiovascular and exercise conditioning. So direct comparisons between outcomes of passive and exertional heat stroke need to be viewed with caution.”

 

Review Summary for Reviewer # 2 

In this manuscript, the authors detail an experiment intended to test whether castration, and therefore the removal of testosterone, from male mice affects the physiological response to EHS. Contrary to previous findings, no substantial differences were observed in the castrated mice compared to the control (EHS) mice. While the sample size is small, I believe the experimental design is adequate, and the conclusions/discussion are in concordance with the reported results. However, I have detailed a few comments that need to be addressed below.

Thank you for your positive view of the manuscript. 

Major Comments

1. The results section currently lacks any display of the p-values that the hypothesis tests are based upon or effect sizes. I would strongly recommend the inclusion of both in the results. In particular, exact/precise p-values (e.g., p = 0.048 not p < 0.05) would be preferrable (even on the figures) since this is considered standard in physiology research https://journals.physiology.org/doi/full/10.1152/advan.00022.2007

Thank you for this reminder. We have now added exact P-values to the text and/or to the figures

a. As for effect sizes, I would recommend eta-squared for the Kruskal-Wallis and the rank-biserial correlation coefficient for the Wilcoxon sum-rank tests. 

b. For the Kruskal-Wallis effect size; �2=(*H* − *k* + 1)/(*n* − *k*), wherein H refers to the test statistic from the Kruskal-Wallis tests, k is the number of groups, and n is the total sample size.

Thank you. Effect sizes have now been included for all groups of measurements in the manuscript, within the figure legends. We used the “eta squared” function for Kruskal Wallis as you described and report it in the figure legend for Fig. 5 A and B. Thank you for this. We also verified the formula and referenced Maciej Tomczak and Ewa Tomczak. “The need to report effect size estimates revisited. An overview of some recommended measures of effect size. Trends in Sport Sciences. 2014; 1(21):19-25.”

c. For the Wilcoxon sum rank effect size see this article by Kirby for a variety of formulaic approaches. https://journals.sagepub.com/doi/full/10.2466/11.IT.3.1

For Wilcoxon Mann Whitney U Test we continued to use G*Power, which has this function for Wilcoxon sum ranks built into it. 

2. In the results, it is unclear which tests are based on an ANOVA/t-test or the non-parametric tests. Statistically speaking, testing for normality can be problematic (suffers from type 1 and 2 error like other statistical tests), and non-parametric only reduce power by a small amount. Therefore, I would suggest just using non-parametric tests throughout the analyses in this paper. This would simplify the interpretation and provide consistency throughout the results.

Clarification has been added to the statistical and analytical approaches. We have now included all statistical tests used for specific plots in the figure legends. Our long-held understanding is that blanket application of nonparametric tests to data with different types of variance, though conservative, increases the possibility of making a type I Error. Therefore, we appreciate your comment but do not think that applying a blanket nonparametric tests is appropriate for all data sets in this report. 

Minor Comments

1. Line 193: it is unclear what statistical programs were utilized for which analysis (JMP or GraphPad) 

The following has been added: 

Statistical testing and graphics were performed using SAS-JMP Pro 15 and/or GraphPad Prism 9.4.

2. Line 198: Kruskal-Wallis is not an “analysis of variance” so calling it an ANOVA is inappropriate here

Thank you, you are correct. This has been removed. Sometimes it is inappropriately referred to as a “nonparametric ANOVA.” For example, in GraphPad Prism, it is lumped within the One Way ANOVA test selection so we tend to make that mistake. 

3. Line 198: I am a tad confused why the Steel-Dwass test was utilized here. This test is rarely used (in comparison to other post-hoc tests). Why use this post-hoc over the more powerful Conover-Imran, Nemenyi, or the Dunn tests?

Kruskal Wallis was used for comparison of three groups in Figure 5A and B, but the overall tests were not significant, so a post hoc was not actually needed for this. That was a mistake in the original paper. This has been removed. 

4. Line 200: I believe I have encountered two typos. First, the Wilcoxon (not Wilcoxin) signed rank test is a one-sample or paired samples test, and I believe the authors may be referring to the Wilcoxon sum rank test (also referred to as the Mann-Whitney U-test or the Wilcoxon-Mann-Whitney sum rank test). Second, the data is not parametric or nonparametric, the *statistical tests* involve the estimation of a parameter or not.

Thank you. This has been corrected and the following has been added: Actually in this submission we consistently used the term Mann Whitney U test throughout the manuscript as that is what it is referred to in our statistical packages.

---

## [Decision Letter · Decision Letter 1]

22 Sep 2022

The impact of castration on physiological responses to exertional heat stroke in mice

PONE-D-22-14934R1

Dear Dr. Clanton,

We’re pleased to inform you that your manuscript has been judged scientifically suitable for publication and will be formally accepted for publication once it meets all outstanding technical requirements.

Kind regards,

William M. Adams

Academic Editor

PLOS ONE

Additional Editor Comments (optional):

Reviewers' comments:

Reviewer's Responses to Questions

**Comments to the Author**

1. If the authors have adequately addressed your comments raised in a previous round of review and you feel that this manuscript is now acceptable for publication, you may indicate that here to bypass the “Comments to the Author” section, enter your conflict of interest statement in the “Confidential to Editor” section, and submit your "Accept" recommendation.

Reviewer #1: All comments have been addressed

Reviewer #2: All comments have been addressed

2. Is the manuscript technically sound, and do the data support the conclusions?

Reviewer #1: Yes

Reviewer #2: Yes

3. Has the statistical analysis been performed appropriately and rigorously? 

Reviewer #1: Yes

Reviewer #2: Yes

4. Have the authors made all data underlying the findings in their manuscript fully available?

Reviewer #1: Yes

Reviewer #2: Yes

5. Is the manuscript presented in an intelligible fashion and written in standard English?

Reviewer #1: Yes

Reviewer #2: Yes

6. Review Comments to the Author

Reviewer #1: (No Response)

Reviewer #2: Thank you for your responses to my review. I have no further comments or concerns about the quality of the manuscript.

7. PLOS authors have the option to publish the peer review history of their article (what does this mean?). If published, this will include your full peer review and any attached files.

Reviewer #1: No

Reviewer #2: **Yes: **Aaron R. Caldwell

---

## [Editor Report · Acceptance letter]

3 Oct 2022

PONE-D-22-14934R1 

The impact of castration on physiological responses to exertional heat stroke in mice 

Dear Dr. Clanton:

I'm pleased to inform you that your manuscript has been deemed suitable for publication in PLOS ONE. Congratulations! Your manuscript is now with our production department. 

Kind regards, 

on behalf of

Dr. William M. Adams 

Academic Editor

PLOS ONE